# Intervention Programmes for First-Episode Psychosis: A Scoping Review

**DOI:** 10.3390/nursrep15010016

**Published:** 2025-01-09

**Authors:** Marta Gouveia, Tânia Morgado, Tiago Costa, Francisco Sampaio, Amorim Rosa, Carlos Sequeira

**Affiliations:** 1Local Health Unit of Viseu Dão-Lafões, 3504-509 Viseu, Portugal; 2Abel Salazar Biomedical Sciences Institute, University of Porto, 4050-313 Porto, Portugal; 3RISE-Health, Nursing School of Porto, 4200-450 Porto, Portugal; tmorgado@gmail.com (T.M.); tiagofilipeoliveiracosta@gmail.com (T.C.); franciscosampaio@esenf.pt (F.S.); carlossequeira@esenf.pt (C.S.); 4Pediatric Hospital of the Local Health Unit of Coimbra, 3000-602 Coimbra, Portugal; 5Health Sciences Research Unit—Nursing (UICISA: E), Nursing School of Coimbra, 3000-232 Coimbra, Portugal; amorim@esenfc.pt; 6Nursing School of Coimbra, 3000-232 Coimbra, Portugal; 7School of Health Sciences, Polytechnic of Leiria, Campus 2, Morro do Lena, Alto do Vieiro, Apartado 4137, 2411-901 Leiria, Portugal; 8Local Health Unit of Gaia e Espinho, 4434-502 Vila Nova de Gaia, Portugal; 9Red Cross Northern Health School, 3720-126 Oliveira de Azeméis, Portugal; 10Nursing School of Porto, 4200-072 Porto, Portugal; 11Research Unit, Nursing School of Porto, 4200-072 Porto, Portugal

**Keywords:** psychosis, early intervention, mental health, psychotic disorders, review

## Abstract

The aim of this scoping review was to map intervention programmes for first-episode psychosis by identifying their characteristics, participants, and specific contexts of implementation. It seems reasonable to suggest that early intervention may be beneficial in improving recovery outcomes and reducing the duration of untreated psychosis (DUP). Despite the expansion of these programmes, there are still some significant variations and barriers to access that need to be addressed. In line with the Joanna Briggs Institute (JBI) methodology and the Participants, Concept, and Context (PCC) framework, this review encompasses studies focusing on individuals grappling with early-stage psychosis and their caregivers across a range of settings, including hospital and community environments. The review identified 47 studies from 2002 to 2023, which revealed a great deal of diversity in programme characteristics and implementation contexts. This reflects a global perspective. The results showed that there is a great deal of variety in the characteristics of the programmes, with interventions ranging from single-component strategies, such as cognitive–behavioural therapy (CBT) and cognitive remediation therapy (CRT), to multicomponent programmes that integrate a number of different approaches, including psychosocial, pharmacological, and family-focused strategies. The objectives included attempts to improve cognitive functioning; enhance coping skills; reduce caregiver burden; and address symptoms such as anxiety, depression, and hallucinations. It is notable that there was considerable variation in the frequency, duration, and follow-up periods of the interventions, with some lasting just three sessions over one month and others spanning five years and 48 sessions. The majority of the programmes were delivered in community or outpatient settings, although there were also examples of hospital- and home-based interventions. These findings highlight the value of early interventions and provide a useful resource for adapting programmes to different social and cultural contexts. It would be beneficial for future research to explore how these interventions can be tailored to diverse settings.

## 1. Introduction

Psychosis is characterized by various signs and symptoms, including changes in thinking and perception [1]. Common features include delusions, hallucinations, mood disturbances, cognitive impairments, and behavioural changes [2,3,4]. The first psychotic episode generally occurs in late adolescence or early adulthood and is defined as the initial experience of such symptoms, lasting for at least one week and significantly impacting the individual’s functioning [3,4].

Early intervention in psychosis is widely recognised as essential for a favourable prognosis [5,6,7]. In recent decades, research in the field of early psychosis intervention and related therapeutic strategies has expanded [8,9], challenging the traditionally negative outlook associated with psychosis [10].

Three key contributions to the field are particularly significant. Firstly, Birchwood’s [11] studies highlight the critical intervention period, identified as occurring within two to five years after symptom onset, during which intervention is most effective and after which the effectiveness of intervention diminishes [1,2,11,12]. Secondly, McGorry’s [13] research on the staging of psychotic illness suggests that appropriate interventions can delay or even prevent progression to more advanced stages [9,13,14,15]. Lastly, the duration of untreated psychosis (DUP) has been recognised as a crucial factor influencing the course of the illness, with shorter DUPs linked to more positive outcomes [16]. This understanding has led to the identification of distinct stages within the disorder, facilitating the development of targeted interventions and a preventive approach [2,12,13,14] to halt progression to more advanced stages [17].

Given the nature of psychosis and developments in the field, early intervention is considered fundamental [14] and more effective than general care [1,5,18,19]. Early intervention requires intervention that is appropriate to the stage of the disease, promotes recovery, and delays or prevents deterioration of the person [20]. It aims to achieve outcomes not only at the clinical level (such as symptom reduction), but also at the personal level in terms of developing a productive and meaningful life [19,21,22]. In this context, the family will play an important role, not only because of the impact that a psychotic break can have on their dynamics [23], but also by actively supporting the individual in the recovery process [24,25] and helping to prevent relapse and social isolation. Their involvement is crucial in addressing the challenges posed by the illness [25].

Since its inception in 1992 in Melbourne, Australia [26], early intervention services have expanded significantly worldwide [1,8,27]. In 2005, the World Health Organization and the International Early Psychosis Association came together in the “Early Psychosis Declaration” to identify the essential components of early intervention services for psychosis, emphasising the need for a broad and eclectic approach [28] to promote recovery (symptomatic and functional) and empower the individual experiencing their first psychotic episode [9,28]. Early intervention must be accessible and comprehensive, involving caregivers. This requires a multidisciplinary approach that incorporates pharmacological and psychosocial interventions to reduce the severity and impact of the disease [1,2,3,22,23]. By implementing appropriate interventions, the aim is to minimise the risk of relapse, enhance the individual’s functioning, and promote recovery as swiftly as possible [12,29,30].

The number of programmes is growing worldwide [21], and the results are encouraging in terms of symptom reduction, overall functioning, quality of life, and relapse reduction [4,31] by reducing DUP, optimizing treatment response, reducing family burden, treating comorbidities such as substance abuse, and preventing disease progression [32]. Currently, there are guidelines with recommendations for the development of these programmes like Orygen [2], Health Service Executive (HSE) [4], and the National Association of State Mental Health Programme Directors (NASMHPD) [33]. Early intervention teams share common goals, including minimising the DUP, developing integrated treatments, and involving families [2,3,4,29,33,34]. However, there is significant variability in how these programmes are implemented [31]. Despite scientific evidence showing that early intervention is more effective than general care [1,18], its dissemination is limited, particularly in low- and middle-income countries [35,36]. In these regions, the development of early intervention programmes is often much slower [33,37] compared to high-income countries, where such programmes are widely established [38].

In this sense, the unequal development of mental health services gives rise to inequalities in access to care, with services often being geographically dispersed [35,36]. Additionally, barriers stemming from health systems or the services themselves reflect inconsistencies in their implementation [39]. This issue is particularly evident in low-income countries, where mental health services are frequently underfunded, and the DUP is often longer, leading to poorer recovery outcomes.

To address these issues, it is essential to create conditions that facilitate the development of services [35,36,37,40]. Improvements in accessibility, equity, and treatment outcomes can only be achieved through systematic implementation within national health systems [32]. Furthermore, ethical considerations must be integrated into the development of care pathways for psychosis, given that access is contingent upon socio-cultural contexts and the structure of health services [41].

Nevertheless, existing early intervention services show significant variation in their delivery models [31], leading to uncertainty about how best to adapt and implement them across different contexts. This variability underscores the need for comprehensive mapping to ensure that services are effectively tailored to the unique demands of diverse healthcare settings, calling for context-specific approaches [42]. The heterogeneity of these services, combined with the complexity of care pathways for first-episode psychosis, often results in a lack of standardised psychometric data, reflecting the diversity of intervention programmes and recovery trajectories [43]. To address these challenges and enhance care models, it is essential to develop fidelity scales that standardise implementation while incorporating quality indicators into their dissemination. This approach will ensure consistent application and maintain the effectiveness of these models across diverse settings [42].

A scoping review of early intervention programmes for psychosis is essential given the considerable variation and lack of standardisation in the way these programmes are described and structured. Although many interventions are documented, information is scattered across different sources, making a comprehensive and coherent understanding of current approaches difficult. This fragmentation hinders a full overview of key programme characteristics, including the type of intervention, facilitators, objectives, frequency of use, context of implementation, and evaluation methods. A scoping review will systematically map the range of existing programmes and provide a broad overview of their structure and implementation in different contexts. Consolidating this information into a single document will facilitate a clearer understanding of the diversity of approaches, enabling future discussion and further research into their adaptability and potential impact in different settings.

A preliminary search of the Cochrane Database of Systematic Reviews, PROSPERO, MEDLINE, and JBI Evidence Synthesis revealed that, although studies on this topic exist, no systematic review has addressed the specificities and scope of early intervention programmes for psychosis. It is therefore crucial to map the characteristics of these programmes to support their development and dissemination. In this context, mental health nurses, with their strong background in evidence-based interventions, play a key role within multidisciplinary teams, applying a holistic approach that takes into account the patient’s social and family context. By increasing their involvement, a more collaborative environment can be fostered, enhancing both the accessibility and comprehensiveness of mental health care. The objective of this scoping review is to map the features of intervention programmes for first-episode psychosis, including their characteristics, participants, and implementation contexts, whether in hospital or community settings.

## 2. Materials and Methods

This review was conducted according to the Joanna Briggs Institute (JBI) methodology for scoping reviews [44]. The Preferred Reporting Items for Systematic Reviews and Meta-Analyses extension for Scoping Reviews (PRISMA-ScR) checklist was used as a structuring matrix [45]. The review protocol was registered in the Open Science Framework on 26 February 2022 [46] and was conducted according to an a priori protocol published in 2023 [47]. OSF Registration Doi: 10.17605/OSF.IO/ZY9QM.

### 2.1. Review Questions

The objective of this study was to map the landscape of early intervention programmes for individuals experiencing a first episode of psychosis. The central review question was:

What early intervention programmes are implemented for service users and their families experiencing first-episode psychosis?

To address this, the following sub-questions were explored:What are the characteristics of these intervention programmes? (e.g., programme name, objectives, frequency, type of intervention, facilitators, evaluation methods, and implementation context)In what contexts are these programmes implemented?Who is the target audience for the intervention programmes (patients and/or family members)?

### 2.2. Inclusion Criteria

This review follows the methodology proposed by the JBI for scoping reviews, utilising the Participants, Concept, and Context (PCC) framework to ensure a comprehensive and structured approach to the collection and analysis of evidence. The key elements of the PCC relevant to this review are outlined below [44,45].

#### 2.2.1. Participants

This review included studies that included people with symptoms associated with the early stages of psychosis. Terms such as “first episode psychosis”, “recent onset psychosis”, “early onset psychosis”, and “early psychosis” may be used to describe participants. The study will not include individuals diagnosed with organic psychosis.

There are no restrictions on the age or gender of participants, with the only inclusion criterion being diagnosis. The study will also include caregivers, defined as first- or second-degree relatives who provide care and support to or live with people with first-episode psychosis. Participants may be either carers, people with first-episode psychosis, or both.

#### 2.2.2. Concept

This review considered studies that explored intervention programmes specifically designed to address first-episode psychosis and early-onset psychosis. These programmes provide a variety of interventions, including psychotherapeutic (e.g., cognitive–behavioural), psychosocial, vocational, and psychoeducational approaches, aimed at assisting both individuals experiencing their first episode of psychosis and their family caregivers.

It is important to note that interventions delivered in a general manner (e.g., “treatment as usual”) during consultations not specifically designated for first-episode psychosis in the early phase are excluded from this review.

#### 2.2.3. Context

This scoping review will consider studies from all countries and settings. This includes both hospital and community environments. There will be no exclusion criteria. We will include studies conducted in both inpatient and outpatient settings, whether psychiatric or non-psychiatric. Interventions delivered by trained healthcare professionals within a clinical intervention context, whether face-to-face, telephone-based, online, or home-based, will be considered.

#### 2.2.4. Types of Sources

Quantitative, qualitative, and multi-method/mixed-method studies were included in the scoping review. Quantitative studies comprise observational research with descriptive, exploratory, and analytical designs. All systematic reviews were included, independently of the types of methods of search used, as well as experimental designs like quasi-experimental studies, randomized controlled trials, and non-randomized controlled trials. Grey or unpublished literature, such as theses, dissertations, reports, government publications, organizational papers, and guidelines, was included. Sources of information had no geographical or cultural limitations and were consistent with the author’s proficiency in English, Portuguese, Spanish, and French. 

### 2.3. Search Strategy

The search strategy was designed to identify relevant studies and reviews, both published and unpublished. We began with an initial search of MEDLINE (PubMed) and CINAHL (EBSCO) to identify terms related to the topic. To develop a comprehensive search strategy for MEDLINE with full-text access via PubMed, we included text words from the titles and abstracts and their index terms (see Appendix A). This search strategy was adapted for each information source to include all identified keywords and index terms according to the inclusion criteria. In addition, we examined the reference lists of the articles included in the review to identify additional relevant articles.

### 2.4. Source of Evidence Selection

The search encompassed a variety of databases, including the Web of Science Core Collection (ISI Web of Knowledge), MEDLINE with Full Text, CINAHL Complete, PsycINFO (accessible through EBSCOhost), Scopus, the Cochrane Library, and JBI Evidence Synthesis. Additionally, efforts to identify unpublished studies involved searching OpenGrey, a European repository, as well as MedNar.

### 2.5. Study Selection

The search results were imported into EndNote vX9 (Clarivate Analytics, Philadelphia, PA, USA), where duplicates were removed. Two independent reviewers assessed the titles and abstracts to ensure they aligned with the inclusion criteria. Articles were selected based on the relevance of their titles and abstracts, including those that lacked an abstract. The reviewers thoroughly analysed any articles that met the inclusion criteria or raised uncertainties.

After this initial assessment, the full texts of the selected citations that complied with the inclusion criteria were reviewed by the two independent reviewers. Any disagreements were resolved through discussion, or, if necessary, a third reviewer was consulted. Full-text citations of eligible studies were uploaded into the JBI System for the Unified Management, Assessment, and Review of Information (JBI SUMARI), developed by the Joanna Briggs Institute in Adelaide, Australia.

Full-text articles that did not meet the inclusion criteria were documented and presented in a PRISMA-ScR flowchart diagram [45]. Authors of articles without access were contacted, and those articles were excluded if access could not be obtained. Due to the volume of articles, those without detailed information on programme characterisation, particularly frequency, were excluded. Articles were retained if they provided, at minimum, general information on the characterisation of the participants (including programme name, intervention objective, frequency (at least two out of four), type of intervention and evaluation). Although some articles did not always provide clear information on intervention facilitators or implementation context, they were still included, even if the information was incomplete.

### 2.6. Data Extraction

The data from the articles selected for the scoping review were extracted by two independent reviewers using a data extraction tool as outlined in the review protocol [47]. The extracted information included comprehensive specifications regarding the intervention programmes examined. In the event of any discrepancies between the reviewers, these were resolved through discussion, and if necessary, a third reviewer was consulted. Furthermore, the authors of the articles were contacted to obtain any missing information, ensuring that additional data were acquired as required.

### 2.7. Data Analysis and Presentation

The text presents the data through visuals, narrative, and tables. It outlines general study information, participant characterisation, programme characterisation, and the implementation context.

General study information includes the author, year of publication, country of origin, type of study, and study objectives. Participant characterisation covers the diagnosis, age of participants, and target group. Programme characterisation encompasses the programme name, intervention objectives, frequency (including the number, duration, and periodicity of sessions, as well as the follow-up period), intervention type (strategy and content), facilitators, and evaluation methods.

The data also address the implementation context, specifying the geographical area, if mentioned, and describing the setting—whether residential, community-based, or outpatient. Where possible, this section also includes the number of participants, indicating whether the intervention was delivered in a group or individual setting.

### 2.8. Study Inclusion

The process of searching and selecting evidence was followed as planned, and the results were synthesized into a PRISMA-ScR (Preferred Reporting Items for Systematic Reviews and Meta-Analyses extension for Scoping Reviews) flow chart, which can be viewed in Figure 1 [48].

## 3. Results Characteristics of Included Studies

### 3.1. General Study Information

This section summarises the general characteristics of the reviewed studies, including the author, publication year, origin, study type, and objectives (see Appendix B). The inclusion of 47 studies reflected a global perspective on early psychosis interventions. Publications spanned from 2002 to 2023. Contributions came from Australia (14.89%) [49,50,51,52,53,54,55], Canada (10.64%) [56,57,58,59,60], and several other countries: China (4.25%) [61,62], Croatia (2.13%) [63], Denmark (4.25%) [64,65], France (2.13%) [66], Germany (2.13%) [67], Iceland (2.13%) [68], India (2.13%) [69], Ireland (6.38%) [70,71,72], Italy (6.38%) [73,74,75], the Netherlands and Belgium (2.13%) [76], Norway (2.13%) [77], Singapore (2.13%) [78], Spain (10.64%) [19,79,80,81,82], Switzerland (2.13%) [83], United Kingdom (17.02%) [84,85,86,87,88,89,90,91], and the United States (6.38%) [31,92,93]. Studies originated from North America (n = 8), Europe (n = 27), Asia (n = 5), and Oceania (n = 7), with no representation from Africa or South America. Various research designs were used. Randomized controlled trials (RCTs) were predominant, including thirty-two studies [49,50,51,53,54,55,56,57,58,61,64,65,68,77,79,80,81,84,85,87,88,93]. Variations included Pragmatic Cluster RCT [73], Multicentre RCT [19], and Pilot RCTs [60,67,70]. Other designs were descriptive studies [52,59,63,66,74,75,78,82,83,92], cross-sectional studies [90], literature reviews [31], prospective controlled trials [76,86], and comparative studies [71]. Experimental designs featured quasi-experimental [69] and waiting list-controlled studies [62]. The objectives varied widely. Some studies explored cognitive interventions, compared therapeutic conditions (e.g., cognitive–behavioural therapy (CBT) combined with treatment as usual (TAU) vs. TAU alone), and evaluated multi-component psychosocial interventions by identifying barriers to feasibility and predictors of treatment effectiveness in first-episode psychosis (FEP) [19,73]. Others piloted programmes like group-based Integrated Cognitive Remediation (ICR) [68] or assessed novel interventions such as the Actissist mobile app [84]. The efficacy of peer-led family support versus traditional family psychoeducation and TAU was also examined [61]. The impact on cognitive functioning, social recovery, depressive symptoms, self-esteem, and quality of life was assessed [77,87,88]. Studies also investigated combining pharmacological and psychological interventions [85,86] and explored cost-effectiveness and satisfaction [50,67]. Additional research evaluated novel psychosocial interventions combining cognitive remediation therapy (CRT) and CBT [70], and RCTs assessed CBT for cannabis cessation [79] and cognitive therapy for suicidal patients [54], while the benefits of cognitive remediation on secondary negative symptoms and social functioning were also explored [93].

### 3.2. Participant Characterisation

The diagnoses considered included FEP (equivalent to recent-onset psychosis, early psychosis, early-onset psychosis, and first episode of schizophrenia spectrum disorder), which corresponded to studies that primarily focused on FEP patients (87.50% of the articles). The remaining 12.50% also referred to FEP patients, but the target group exclusively comprised the caregivers [50,52,55,62,69,90].

The age of the patients ranged from 12 [74] to 65 years [76,77,89], with different intervals depending on the article. The most common age range was 18–35 years, with 10.42% of articles, followed by 18–45 years, with 8.33%. Twenty-four articles dealt with the age group below 18 years [31,49,51,53,54,56,59,62,66,67,70,74,75,76,78,79,81,84,85,86,87,88]. Some articles did not specify the age (e.g., [63,64,71]), and three articles included the age group of over 65 years [76,77,89].

Of the selected articles, twenty-three included interventions only for patients, eighteen included interventions for both carers and patients, and six included interventions only for carers [50,52,55,62,69,90] (see Appendix C).

### 3.3. Programme Characterization

The included studies presented different interventions in terms of their objectives, frequency, intervention types (strategies and content), facilitators, and evaluations (see Appendix D).

#### 3.3.1. Programme Name/Intervention Objective

Given the high number of articles, an effort was made to group them into single-component interventions (e.g., CBT, computerised interventions, cognitive remediation, psychoeducation) or those that integrated multiple components (e.g., CBT + CM + psychoeducation). This categorisation was adopted for two primary reasons: to facilitate the organisation and analysis of the articles and to highlight the programmes’ characteristics. The classification provides a clear framework for presenting programmes with varying levels of complexity, which allows us to underscore the depth of focused interventions and the breadth of integrated programmes without overwhelming the reader with unstructured details. The included studies encompassed different psychosocial interventions, reflecting a wide array of cognitive, behavioural, and social approaches tailored to varying objectives and characteristics. It is acknowledged that some single-component interventions may have been part of broader programmes; however, they were analysed independently when the study’s primary focus was on a single component, as specified in the original articles.

Objectives: The core aim of each programme is to address a specific aspect of psychosis, such as improving cognitive function; enhancing coping skills; alleviating caregiver burden; or reducing symptoms like anxiety, depression, or hallucinations. These objectives are the intended outcomes of the intervention, guiding its design and scope.

Appendix E provides a brief overview of the different programmes, including their names (if assigned), intervention objectives, and whether they are single- or multi-component. Note that there may be selected articles in which only a single isolated intervention is analysed, which might be part of a broader programme (e.g., [54]).

##### Single-Component

Single-component interventions primarily focus on cognitive and cognitive–behavioural therapies. CBT is a prominent feature of numerous studies [51,57,67,71,76,77,79], addressing a range of objectives, including the reduction in both positive and negative symptoms, enhancement of overall functioning, support for cannabis cessation [79], and management of social anxiety [57].

In the realm of cognitive interventions, CRT is a central focus, with several studies indicating its efficacy in improving cognitive function and supporting functional recovery [78,86,87,93]. The objective of CRT is to enhance cognitive abilities and provide strategies to manage cognitive deficits. In addition to CRT, compensatory cognitive training (CCT) is a key element of this approach, intending to develop new cognitive habits to adapt to impairments [60].

Several studies have explored the potential of recovery-focused interventions, including bibliotherapy and problem-solving techniques. The use of bibliotherapy has been demonstrated to provide support to caregivers and alleviate psychological distress [50,55], and psychoeducation has been shown to assist patients and families in the management of early-onset psychosis [81]. Furthermore, the Method of Levels therapy [89] seeks to resolve goal conflicts and enhance self-management, while the Cognitive Recovery Intervention [88] is designed to reduce trauma symptoms and boost self-esteem or detect and monitor suicide-risk patients [54].

Other interventions include mindfulness-based social cognition training [80], which promotes a non-judgemental approach to interpersonal relationships, and psychoeducation [52,62,90], which enhances carers’ understanding of psychosis. A noteworthy innovation is a computerised approach to managing psychosis in real time [84].

##### Multicomponent

Multicomponent interventions integrate multiple therapeutic strategies to offer a comprehensive approach to early psychosis management. Programmes such as Cognitive Adaptation Training (CAT) and Action-Based Cognitive Remediation (ABCR) combine home-based supports with computerised cognitive exercises to address cognitive and motivational issues [56]. The Cognitive Remediation and Social Recovery in Early Psychosis (CReSt-R) programme merges CRT with social recovery therapy to improve both cognitive and social functioning [70].

Integrated approaches are significant in early psychosis management. The NEUROCOM programme combines cognitive remediation with OPUS treatment, which includes social skills training, patient psychoeducation, and family interventions [64]. Similarly, the cognitively oriented psychotherapy for early psychosis—COPE—programme integrates cognitive/behavioural therapy with psychoeducation and case management to facilitate patient adjustment and prevent secondary morbidity [49]. The NAVIGATE programme offers a comprehensive package including family education, individual resiliency training, and supported employment and education [92]. The Parma-Early Psychosis (Pr-EP) programme also includes a multi-component psychosocial intervention [74].

Family-focused interventions are crucial, with programmes like the Family-Led Mutual Support Group (FMSG) combining family psychoeducation with support groups to enhance family functioning and reduce rehospitalisation rates [61]. The Integrated Treatment Programme includes assertive community treatment (ACT), social skills training, and multifamily groups to address psychotic and disorganised symptoms [65]. The comprehensive therapeutic programme (CTP) utilises a range of therapies, including psychodynamic group psychotherapy, cognitive–behavioural workshops, and occupational therapy, to achieve remission and recovery [63].

Extensive programmes also play a vital role. The DETECT initiative combines CBT with occupational therapy and a Carer Education Programme to address early-phase psychosis and improve care [72]. The PEPP programme integrates cognitive skills training, family support, and individual therapeutic interventions to prevent relapse and support recovery [59]. The POTENTIAL programme employs a multidisciplinary approach, including individual and group therapies, to prevent chronic mental illness [31].

The combination of CBT and psychoeducation targets clinical improvement through normalising information, problem-solving, and relapse prevention [85] or enhancing functioning, treatment adherence, and illness awareness [19]. The Integrated Need-Adapted Treatment programme focuses on individual psychotherapy, group therapy, and improving treatment adherence [82]. Combining CR with CBT aims to enhance cognitive skills and facilitate effective symptom management [91]. The psychoeducational/psychosocial management approach targets social support and reduces family burden [69].

The EPPIC programme employs a multi-modal therapeutic approach, including relapse prevention and family-based CBT to prevent relapse after a first episode of psychosis [53]. The NEAR programme (Neurocognitive Educational Approach to Remediation) incorporates cognitive remediation, CCT, and social cognition and interaction training (SCIT) [68]. The Re-Arms programme combines CBT, case management (CM), and psychoeducation to improve overall treatment outcomes [75]. Additionally, the CBT plus psychoeducation approach [66] targets a reduction in psychotic symptoms and aims for a greater improvement in overall functioning, while ACT combined with case management [83] enhances continuity of care and reduces inpatient admissions.

Moreover, group treatments such as CBT for psychosis (CBTp) and symptom management (SM) programmes aim to enhance multiple protective factors, including skills, social competencies, family and social support, adaptive strategies, self-esteem, stress management, and medication compliance [58]. The AVEC component empowers families to support each other and provides information on various aspects of psychosis, contributing to the holistic approach to treatment and support.

The combination of CBT, case management, and family intervention for psychosis (FIP) enhances functioning, treatment adherence, and understanding of the condition, with a more substantial reduction in depressive, negative, and general psychotic symptoms following treatment [73].

#### 3.3.2. Frequency

The reviewed articles present a wide range of intervention programmes for psychosis, with considerable variation in the number of sessions (NS), treatment duration (TD), session frequency (FS), and follow-up (FU). The number of sessions varies significantly, with some programmes offering as few as 3 sessions [90], while others have up to 48 sessions [93]. Treatment durations also vary greatly, ranging from one month [52] to five years [82]. The frequency of sessions is similarly varied, ranging from daily [84] to fortnightly [19] or even monthly [80]. Follow-up periods are also inconsistent, with some studies having no follow-up (e.g., [52,68,78]), while others extend for up to 24 months or more after treatment [67].

Due to the wide heterogeneity of values for NS, TD, FS, and FU, the characteristics were described with a presentation of their amplitudes (minimum and maximum values) to illustrate the variation in the data, when possible. It is important to note that several articles did not report all relevant details, such as the duration of the sessions (e.g., [53,76]) or the exact follow-up periods (e.g., [66]), highlighting the need for more consistent reporting in the literature. In addition, a subset of articles (e.g., [63,89,92]) describes comprehensive interventions that are highly adaptable, with the number and frequency of sessions tailored to individual needs. These flexible programmes are common in cognitive–behavioural (CBT) and metacognitive training settings, emphasising personalised care to address both clinical and psychosocial concerns.

Given the large number of articles and the variation in the reported data, this summary provides an overview at a global level. More detailed information can be found in Appendix D.

#### 3.3.3. Intervention Type—Strategy/Content

Intervention programmes for psychosis include a wide range of strategies to support people at different stages of their illness.

Strategies and Contents: These refer to the specific therapeutic approaches and tools used to achieve the intervention’s objectives. For instance, CBTp focuses on the management of psychotic symptoms through structured phases, including goal setting and relapse prevention (e.g., [67]). Alongside CBTp, family interventions provide psychoeducation and support to families, aiming to enhance their ability to effectively manage psychosis-related challenges and foster resilience in the home environment (e.g., [61,69,92]). Comprehensive models, such as the NAVIGATE programme, combine various elements, including family education programmes (FEP), individual resilience training (IRT), supported employment and education (SEE), and individual medication management, to create tailored treatment plans [92]. These strategies promote recovery by integrating personal, social, and vocational support to address the multifaceted needs of patients. Cognitive remediation approaches like SCIT and CCT are aimed at improving cognitive functioning and social skills, directly addressing the cognitive deficits often associated with psychosis and complementing other therapeutic strategies such as CBTp or family support interventions [68].

The combination of psychoeducation and caregiver support programmes provides essential information and practical strategies to caregivers, which are critical for managing the long-term effects of psychosis on both the individual and their loved ones [69,72]. These strategies aim to reduce caregiver burden and improve overall care. Given the number of articles, this summary provides a general overview of strategies. More detailed information can be found in Appendix D.

#### 3.3.4. Intervention Facilitators

Interventions for psychosis involve a wide range of professionals, each with specific roles and different qualifications. Clinical psychologists are often associated with CBT and receive regular supervision to ensure the effectiveness of interventions (e.g., [63,67,77]). Psychiatrists play a crucial role in medication management and therapeutic support, collaborating with the team (e.g., [66,72,74]). Mental health nurses are particularly prominent in community and family therapy interventions. They provide psychoeducation, facilitate support groups, and manage cases, although they usually do not deliver CBT directly (e.g., [52,69,75,89]). Occupational therapists and social workers also play an important role in cognitive rehabilitation and psychosocial support (e.g., [61,64,80]).

Articles focusing on single-component interventions focus on specific techniques requiring specific training for each approach. In contrast, multi-component interventions are a combination of different methods and reflect a broader integration of professionals. Many articles do not fully describe the training or role of facilitators, but when they are mentioned, they are health professionals, often with specific experiences of psychosis. Appendix D provides more information about these practices and facilitator training.

#### 3.3.5. Evaluation

The selected articles yielded a considerable number of scales for the assessment of variables about mental health. Given the considerable diversity and quantity of instruments encountered, grouping these scales into categories was deemed appropriate, thus facilitating data analysis and interpretation. The categories were formed based on the key areas of assessment, which included: psychiatric symptom evaluation, functional assessment, cognitive assessment, well-being and quality of life assessment, and family and social support assessment (see Appendix F).

The category most frequently utilised was that of psychiatric symptom evaluation. The Positive and Negative Syndrome Scale (PANSS) and the Brief Psychiatric Rating Scale (BPRS) were identified as the most recurrent scales within this category. These instruments were used extensively for the assessment of symptom presence and severity. In the category of functional evaluation, instruments such as the Global Assessment of Functioning (GAF) and the Social and Occupational Functioning Assessment Scale (SOFAS) were employed with considerable frequency. These tools assess the impact of mental disorders on people’s daily lives, providing an integrated view of social and occupational functioning.

In the context of cognitive assessment, the Wechsler Adult Intelligence Scale (WAIS) and the Cambridge Neuropsychological Test Automated Battery (CANTAB) were frequently referenced. These scales are used to assess patients’ cognitive abilities. This is an area of growing interest in psychiatry. Moreover, the domain of well-being and quality of life was a significant area of emphasis, with the EuroQol 5-Dimensions 5-Levels (EQ-5D-5L) and the World Health Organization Quality of Life (WHOQOL-Bref) being the most frequently referenced instruments. These instruments are employed to ascertain the patients’ perceptions of their quality of life. In several articles, the use of any scales was not specified. Moreover, several studies opted to employ a combination of scales to gain a more comprehensive understanding of the subject matter, thereby highlighting the intricate nature of the topic under investigation (See Appendix D).

### 3.4. Implementation Context

In examining the implementation contexts of various interventions, many studies were conducted in urban settings [50,51,61,68,71,72,76,85,88,93]. Research in semi-rural and rural areas comprised five studies [54,57,73,74,84]. Additionally, a substantial number of studies did not specify their implementation contexts [19,31,49,52,53,55,56,58,59,60,62,63,64,65,66,69,70,75,77,78,79,80,81,82,83,85,86,87,89,90,91,92].

#### 3.4.1. Setting

The analysis identified three primary settings for interventions. Inpatient settings involve interventions conducted within hospitals or residential facilities [59,69,83,87]. Home-based settings deliver interventions within patients’ homes [49,50,51,56,76,84,91]. Community-based and outpatient settings utilise local resources and encompass interventions conducted in clinics, community centres, or other non-residential environments [19,31,52,53,54,55,57,58,59,60,61,62,63,64,65,66,67,68,71,72,74,75,77,78,79,80,81,82,83,85,86,87,88,89,90,92,93]. Notably, only one review addressed interventions in both inpatient and outpatient settings [83].

#### 3.4.2. Individual/Group Intervention

Additionally, interventions varied in their session formats. Some were exclusively individual [86,89], while others included both individual and group sessions [50,59,81]. Group sizes ranged from small (4 participants) to larger groups (up to 15 participants) [78,80,86]. Community-based and outpatient interventions often featured group formats but did not always specify the number of participants [52,90]. Overall, while individual interventions were predominant, several studies incorporated both individual and group sessions, tailored to the needs of participants and the specific setting.

Individual interventions involve therapeutic approaches delivered on a one-on-one basis between the therapist and the patient [19,49,51,53,54,55,56,58,59,62,63,66,67,70,75,76,77,78,79,80,81,83,84,85,86,87,88,89,93]. Group interventions describe therapeutic approaches delivered to multiple participants simultaneously in a group setting [52,58,60,61,62,63,65,69,71,72,80,86,87,90]. Mixed individual and group interventions combine elements of both individual and group therapy [31,50,51,53,58,59,61,62,63,64,65,68,73,74,81,83,88]. Several articles do not specify whether the intervention is individual or group-based [19,31,52,53,54,55,56,57,59,63,64,66,67,74,75,77,78,79,80,81,82,83,88,89,90,91,92]. For detailed information, see Appendix D).

## 4. Discussion

The scoping review provides a detailed overview of early intervention programmes for first-episode psychosis, revealing considerable diversity in their characteristics, participants, and delivery contexts. To the authors’ knowledge, this is the first review to map programmes along these lines. Of the 47 articles included, there was variation in terms of the structure and type of study. On the other hand, their focus—intervention programmes—also varied in terms of objectives, types of intervention, and implementation strategies. They therefore reflect the inherent complexity of psychosis [94] and the diverse needs of individuals and their families, as highlighted in studies on the staging models of psychosis [13] and on the needs of patients and their families [32].

The findings confirm the importance of a comprehensive and tailored approach to psychosis treatment, aligning with previous research advocating for interventions designed to meet specific manifestations of the illness [13,95]. There is widespread agreement in the literature on the superiority of intensive, team-based interventions for FEP compared to TAU [18,94]. Throughout our review, we can see that, of the 23 articles identified as single-component, 16 are integrated into broader therapeutic programmes where, in addition to the intervention under study, other specific psychotherapeutic or psychosocial interventions for FEP may already be available [50,51,54,55,57,60,62,78,80,86,88,89,90,93]. This finding is supported by recent research highlighting the effectiveness of integrated approaches in treating complex psychopathology. The analysis by Williams et al. reinforces the effectiveness of integrated techniques, showing that models that combine a comprehensive package of treatments are more effective in optimising outcomes and meeting the specific needs of patients in the long term, suggesting that, although some interventions may appear isolated, they are often part of more complex and multifaceted therapeutic contexts [94]. Examples include PSBI (problem-solving therapy) [50,55], which is part of the specialist FEP centres Orygen Youth Health (OYH) and the Recovery and Prevention of Psychosis Service (RAPPS), or even CRT for psychosis, which is part of the NHS early intervention services in the UK [86]; CRT is part of the Early Psychosis Intervention Programme (EPIP) in Singapore [78].

Among the selected articles, many included interventions aimed at carers. These family interventions were frequently associated with reported benefits, such as a reduction in caregiver burden and positive effects over time [96]. Additionally, the studies indicate that these interventions are linked to potential improvements in outcomes such as reduced relapse rates, shorter hospital stays, and psychotic symptoms, as well as improved functionality in patients with first-episode psychosis up to 24 months following the intervention [97]. It has therefore been suggested that these interventions should be incorporated into mental health services, as they offer benefits such as reducing the burden on carers and improving their emotional well-being, as well as helping them to cope with challenges such as the uncertainty and stigma associated with caring [98]. Despite the emphasis in the literature on the need for a holistic and integrated approach [98], in our analysis, 24 articles focused on carers, which is consistent with Claxton et al. [99], who suggest that carers’ needs and the emotional impact of caring remain areas that are often neglected.

In terms of the age groups covered, they vary, but are mainly focused on adolescents and young adults, with varying ranges, which is in line with McGorry et al. and Fusar-Poli et al., who inform us about a youth-oriented intervention [32,100]. In our case, the age range of 14–35 covers the largest number of studies. Some articles cover a wider population, including ages as young as twelve or as old as sixty-five. This reflects the importance of early intervention, particularly during the transition from adolescence to adulthood, and is in line with Addington, who tells us that early intervention for psychosis should take place from early adolescence to adulthood, with a gradual decline up to the age of sixty [101]. Others do not specify an age, suggesting a more comprehensive or generalist approach to intervention. Overall, the diversity of age groups highlights the need for intervention strategies adapted to different phases of life [102]. The health benefits also exist for later implementation, returning to the example of the UK, where guidelines have extended the age of eligibility to sixty-five [103].

In terms of interventions, CBT, CRT, and psychoeducational interventions stand out. CBT was widely used, appearing alone in several articles [51,57,67,71,76,77,79] as well as in combination with other modalities such as cognitive remediation, psychoeducation, and case management [19,49,53,58,66,72,73,74,75,85,91]. Combinations including elements of CCT and mindfulness-based therapies have also been identified [68,80]. This diversity reflects growing evidence in the literature suggesting that integrated approaches are frequently associated with better management of psychotic symptoms and promotion of functional recovery, as they address both cognitive and behavioural aspects [98,104].

These data are consistent with Gergov et al., who report that there are benefits to offering cognitive, behavioural, or CBT and CRT psychotherapeutic interventions to patients, to carers, or in group settings, especially when psychoeducational elements are included [105]. In addition, multi-component programmes such as NAVIGATE [92] and POTENTIAL [31] reinforce the importance of comprehensive approaches that integrate multiple therapeutic techniques and synergistically address the diverse needs of patients.

According to Breitborde et al., optimising interventions in the first psychotic episode also involves a synergistic combination of interventions [106]. In this sense, psychoeducation is a component of evidence-based intervention, as is case management with a comprehensive approach to the patient’s needs [101,107]. Interventions such as OPUS [65], which combines case management with social skills training and multi-family groups, are examples of this integrated approach. They aim for both clinical stabilisation and social reintegration. On the other hand, CBT is also effective in reducing positive symptoms, while family interventions are effective in preventing relapse [108]. Williams et al. suggest that psychological interventions and case management, together with pharmacotherapy, are the central components of services for early psychosis to achieve sustained clinical benefit [94].

These findings suggest that the diversity of interventions reflects the complexity of the treatment of psychosis. To improve outcomes in a complex and heterogeneous syndrome such as psychosis, it is necessary to employ complex intervention models globally [32]. This comprehensive focus is essential given that early intervention services for psychosis are effective in the broader context of mental health care, which is supported by various guidelines such as the Australian Clinical Guidelines for Early Psychosis [2]. At the educational level, interventions aim to improve knowledge about the disease and its treatment, both for patients and their carers, to facilitate treatment adherence and promote a supportive environment [19,31,52,54,58,59,62,63,72,73,74,81,82,83,85,89,90,91,92]. Finally, at the social level, the aim is to strengthen family and social relationships, improve communication and mutual support, and reduce the emotionality expressed in family interactions, which is crucial for successful recovery and maintenance of mental health [50,55,58,59,61,68,69,80,82,92].

In the analysis of the objectives of the interventions, there is an underlying multidimensional basis that aims to promote improvements at different levels. There is variability between programmes and types of interventions. This is consistent with findings in the literature that point to the existence of multiple approaches [18,107]. Our findings were consistent with these findings, with broad aims. Clinically, interventions aim to reduce positive and negative psychotic symptoms, improve cognitive function, and prevent relapse [19,31,49,51,53,56,57,58,59,60,61,63,64,65,66,67,68,70,71,72,73,74,75,76,77,78,79,82,83,84,85,86,87,89,91,92,93]. Functionally, the focus is on restoring social, occupational, and educational skills to enable meaningful reintegration into society [19,31,49,51,57,59,64,66,68,70,71,72,73,74,75,76,77,79,84,91,92]. Psychological aims include supporting emotional well-being, reducing psychological distress for both patients and carers, and promoting a more stable and less stressful family environment [31,49,50,54,55,56,57,59,67,69,72,73,74,75,76,77,80,81,82,83,88,89,90,92]. These goals highlight the importance of a holistic approach to recovery, which goes beyond symptom reduction and promotes an overall improvement in the quality of life and functioning of patients and their families. Fusar-Poli et al. refer to secondary prevention by highlighting the fact that services promote the reduction in SUD. In terms of interventions, they are based on improving the response to treatment, with improvements in well-being, functioning, and social skills, and reducing the burden on the family. They also promote the treatment of comorbid substance use and the prevention of disease progression [32]. However, although the objectives of the interventions imply health benefits, it is still unclear how they should be developed to enable their long-term maintenance [109].

When analysing the frequency and duration of interventions, the programmes varied considerably in terms of the number of sessions, length of treatment, duration of sessions, frequency of interventions, and follow-up periods, reflecting the complexity of treating psychotic illnesses and the need for individualised approaches. Our findings are consistent with Birchwood’s studies, which indicate the existence of a critical period of intervention, a period that can last up to 5 years, during which there is a possibility of achieving more fruitful results [11]. Thus, the existence of variability in frequency is also in line with Chan et al., who argue that interventions should be culturally adapted and tailored to the individual needs of patients, highlighting the importance of a personalised approach to treatment [110].

The variation in follow-up times highlights the importance of ongoing monitoring to assess the long-term effectiveness of interventions and ensure the sustainability of therapeutic gains. However, the variation in follow-up times reflects the lack of a standardised protocol and points to the need for future studies to explore the long-term effectiveness of interventions in order to better guide clinical practice [94].

Although early intervention in psychosis has positive short-term outcomes, there is still uncertainty about the maintenance of these benefits after five years of treatment [18,111]. These findings are in line with Hegelstad, who confirmed the benefits of early intervention but highlighted the need to identify strategies to maintain these benefits in the long term [112]. Favourable outcomes are not always maintained in the long term [111], especially when a transition to treatment as usual occurs [113].

Interventions are delivered by a wide range of professionals, including clinical psychologists, psychiatrists, mental health nurses, occupational therapists, and social workers. The training and supervision of these professionals vary widely, from specialist training in CBTp [49,58] to the implementation of intervention models such as NEAR [93,114] or CRT [86].

Most interventions include regular supervision, feedback sessions, and, in some cases, monitoring of adherence to the protocol (e.g., [85]), which has been associated with improved treatment efficacy [115]. It also suggests that the quality of the intervention may be directly related to the training and ongoing supervision provided to therapists. Furthermore, the presence of a multi-professional team, as in the ReARMS and OPUS programmes (e.g., [65,75]), highlights the importance of a collaborative and holistic approach to the treatment of psychosis. Nevertheless, there is considerable variation in the training of intervention facilitators. Some studies report intensive and specific training (e.g., [58]), while others mention minimal training or do not specify training criteria (e.g., [79]).

Regarding assessment, the results show that a wide variety of assessment instruments are used in early intervention programmes for psychosis. The diversity of scales, covering areas such as family functioning, quality of life, cognitive assessment, psychiatric symptoms, social and occupational functioning, anxiety, depression, self-esteem, and illness awareness, highlights the complexity of psychosis treatment [94]. The multiplicity of domains assessed highlights the need for a multidisciplinary and personalised approach to the quality of care provided [116]. However, the variability of the instruments used may also indicate a lack of standardisation, which can make it difficult to compare results between different studies and programmes [116]. Thus, these results suggest the importance of continuing to explore which tools offer greater sensitivity and specificity to assess the various dimensions of the psychotic experience, promoting a better understanding of the illness and interventions in its trajectory, ultimately improving treatment outcomes and guiding clinical practice. Regarding implementation contexts, it has been found that inpatient environments can be more disruptive in various psychosocial aspects, and, in the sense of recovery, it is advocated that inpatient stays occur as a last measure and for the shortest time necessary, with a smooth transition to care in the community (whether outpatient or community) (e.g., [59,83,87,117]), which is in line with the majority of the articles selected where the implementation context, although varied, takes place in outpatient settings (e.g., [58,64]), at home (e.g., [49,51]), and in the community (e.g., [66,74]).

Even so, according to Siebert and colleagues, specialised inpatient services can be an asset to effective global intervention in the event of the need for hospitalisation [118], and communication between inpatient services and subsequent outpatient follow-up is an indicator of quality [108].

Several limitations were identified in this review. Access to some full-text articles was not always possible, potentially excluding relevant studies. Only studies published in English, Portuguese, Spanish, and French were included, as these were the languages spoken by the authors. This ensured the quality of the review, but may have limited its scope. Additionally, while some studies lacked detailed information on programme characteristics, those with sufficient data to meet the review’s objectives were included. Finally, the decision to exclude interventions not specifically designed for early psychosis may have omitted broader approaches, though this was a necessary methodological choice.

## 5. Conclusions and Implications

This scoping review highlights the importance of early intervention in psychosis and maps the extensive existing research on appropriate interventions in this area. By identifying the characteristics of current programmes, we reveal a diversity of approaches and variability in implementation strategies. This mapping provides a valuable resource for adapting programmes to diverse political, social, and cultural contexts.

The implications of this mapping are considerable. It provides a solid basis for researchers and health professionals to explore interventions that can improve access to mental health care. Furthermore, this review contributes to the development of a specific early intervention programme designed specifically for mental health nurses. Such a programme would enhance their role in multidisciplinary teams and equip them with the tools to provide timely, patient-centred care. While the findings of this review emphasize the importance of multidisciplinary collaboration, they also highlight the unique contributions that mental health nurses can make in supporting holistic and inclusive care. To improve clinical practice and ensure high-quality mental health care, future research should focus on exploring the effectiveness of the mapped interventions and how they can be adapted to different contexts.

## Figures and Tables

**Figure 1 nursrep-15-00016-f001:**
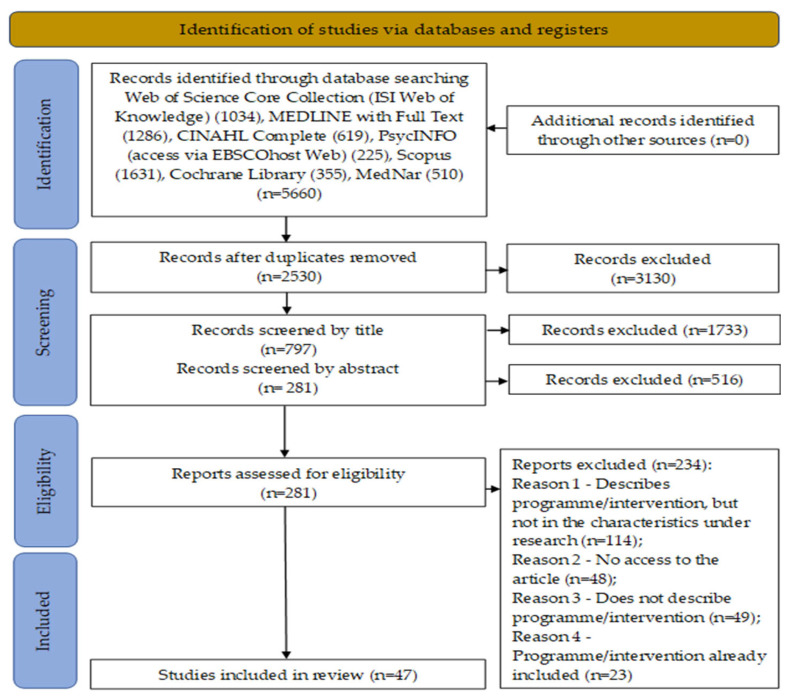
PRISMA flowchart of the screening and assessment process.

## Data Availability

Not applicable.

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
