# Peer review of "Intervention Programmes for First-Episode Psychosis: A Scoping Review"

_nursrep, 2025, doi:10.3390/nursrep15010016_

Round 1

Reviewer 1 Report

Comments and Suggestions for Authors

RE: Manuscript Nursrep-3262226: Intervention Programmes for First-Episode Psychosis: A Scoping Review

COMMENTS FOR THE AUTHOR:

I enjoyed reading the manuscript, which examines existing literature to map intervention programs for first-episode psychosis. This study addresses an important and timely topic area and addresses a gap in research as it relates to a specific population. The study is methodologically strong and the methodology and analysis is thoroughly outlined.

My main concern of the manuscript pertains to writing mechanics, specifically organization of paragraphs throughout the manuscript and formatting with tables. I believe this study is well designed, extremely important and informative, and I was excited to review it. I’ve listed specific feedback based on the line number below.

Introduction

Line 43: Clarify what is meant by this field.

Lines 54-56: Combine this sentence with the paragraph above.

Lines 85-86: Provide the term and then the acronym. For example, Health Service Executive (HSE) and National Association of State Mental Health Programme Directors (NASMHPD).

Lines 95-104: Combine into one paragraph for clarity.

Line 103: Since the acronym was already introduced and used, can just use DUP without defining the term.

Lines 119-122: Combine with paragraph above.

Lines 131-135: Combine with paragraph above.

Materials and Methods

Line 150: Define JBI. For example, Joanna Briggs Institute (JBI) methodology.

Line 169: Can use JBI, since it is described in line 150.   

Line 210: Change was to were.

Line 243: Citation format is inconsistent with other citations in the document.

Line 248: Spell out 4 for consistency.

Figure 1: In identification box, there needs to be spaces after MEDLINE with Full Text (1286), Scopus (1631), Cochrane Library (355), and MedNar (510). There are inconsistencies throughout the Figure with spacing when reporting results. For example, (n = 47) and (n=23). Revise Figure for consistency.

Results

Line 285: Replace forty-seven with 47.

Line 332: Punctuation after the parenthesis. For example, “…] (see Appendix C).”

Line 539: Instead of spelling out fifteen, provide the number (i.e., 15) for consistency in the sentence.

Discussion

Line 558: Replace forty-seven with 47.

Line 568: Can use the acronym since it was previously described in the manuscript.

Lines 591-597: Move information up into previous paragraph.

Line 598: Begin new paragraph.

Lines 638-644: Move this paragraph up to the previous paragraph.

Lines 694-709: Combine into one paragraph as this information pertains to interventions.

Line 701: Appears information is missing from the sentence.

Lines 729-732: Move to above paragraph.

Line 736: Consider changing the word reviewers to authors.  

Appendix B

Jackson citation: Change comma to period for consistency.

Vidarsdottir citation: Change comma to period after al. for consistency.

Chien citation: Remove period after et. and add space.

Krarup Get citation: I think it should read Krarup Get T, et al. to match citation format in the Table.

Reininghaus citation: Missing s on Netherlands.

Overall feedback: Throughout the table in the objectives section, sometimes acronyms were used and sometimes they were not (sometimes they were described and sometimes they were not). For clarify and ease for the reader, I suggest providing a footnote of the acronyms used (i.e., CBT) to avoid having to describe each of them throughout the table.

Appendix C

[52] Early psychosis: A space is missing between 34 and years.

[53]: Remove extra spacing between [53] and FEP.

[54]: Remove space between / and family.

[56]: For consistency, lowercase family.

[59]: Remove extra space between ; and (.

[63]: Remove extra space between patient and /.

[66]: Add space between 35 and years.

[66]: Add space between 45 and years.

[75]: Remove extra space between risk and for.

[81]: Remove extra space between ; and [.

[86]: Lower case family for consistency.

[89]: Lower case family for consistency.

[90]: Remove extra space between / and family.

[94]: Remove extra space between [ and FEP.

Overall feedback: A footnote explaining the elements of the table would help clarify the information for the reader. Additionally, it would help the reader understand the table if the table header format mirrored how the information was displayed. For example, Diagnosis/Age/Threat could read as Early psychosis/17-34 years/Patient OR Diagnosis; Age; Threat, which matches the current format.

Appendix D

Overall feedback: The content provided is detailed and thorough. However, some minor revisions are needed.

In the implementation context column, ensure the first word after the bullet is capitalized for consistency.

Appendix E

Overall feedback: The content provided is detailed and thorough. However, attention to formatting and how the information is presented is needed.

For consistency, lower case name and remove uni (since multicomponent is used throughout the manuscript).

Throughout the table, some of the modalities are capitalized and some are not, and some are explained with a dash and some with a parenthesis. For consistency purposes, please select how you want to visually display this information and make the necessary revisions throughout the table (right column).

I was not able to identify why some of the modalities were bold and some were not. Please clarify. Perhaps a footnote explaining would help clarify for the reader.

In the intervention objective descriptions, some of the descriptions end with a period and some do not. For consistency, I suggest having all descriptions end with a period.

Appendix F

Overall feedback: Information is presented in a clear and concise manner. However, there are some errors with formatting. For example, the Services & Resources and Other categories are missing the brackets around the article numbers. In the Services & Resources category, the EPQ alignment is off. Also, capitalization of words for some of the scales is missing (i.e., Level of expressed emotion should read as Level of Expressed Emotion (LEE)). Please revise table for consistency.

Author Response

Thank you very much for your thoughtful comments and suggestions. We greatly appreciate the time and effort you have taken to review our manuscript. Your feedback has been invaluable in improving the quality of our work.
Please find our detailed responses below, along with the corresponding revisions and corrections highlighted in the re-submitted files using track changes.
Thank you once again for your thorough review and constructive insights

1. Point-by-point response to Comments and Suggestions for Authors

Introduction
Line 43: Clarify what is meant by this field. 
ANSWER: We thank the reviewer for this suggestion. 
ACTION TAKEN
1 - The following sentence has been improved:
“In recent decades, research in the field of early psychosis intervention and related therapeutic strategies has expanded [8-9], challenging the traditionally negative outlook associated with psychosis [10].”

Lines 54-56: Combine this sentence with the paragraph above. 
ANSWER: Thank you for your suggestion. 
ACTION TAKEN
Following the reviewer's guidance, the two paragraphs have been connected to enhance the flow of the text, ensuring a smoother transition between ideas without altering the original content.

Lines 85-86: Provide the term and then the acronym. For example, Health Service Executive (HSE) and National Association of State Mental Health Programme Directors (NASMHPD). 
ANSWER: Thank you for pointing that out.
ACTION TAKEN
Following the reviewer's suggestion, the change was made as proposed.

Lines 95-104: Combine into one paragraph for clarity. 
Line 103: Since the acronym was already introduced and used, can just use DUP without defining the term. 
ANSWER: We appreciate your insightful comment.
ACTION TAKEN
The two paragraphs were combined while maintaining their original meaning, and the full written form of DUP was omitted, leaving only the acronym.

“In this sense, the unequal development of mental health services gives rise to inequities in access to care, with services often being geographically dispersed [38,39]. Additionally, barriers stemming from health systems or the services themselves reflect in-consistencies in their implementation [42]. This issue is particularly evident in low-income countries, where mental health services are frequently underfunded, and the DUP is often longer, leading to poorer recovery outcomes.”

Lines 119-122: Combine with the paragraph above. 
Lines 131-135: Combine with the paragraph above.
ANSWER: Thank you for your suggestion. 
ACTION TAKEN: Both situations were addressed. The paragraphs were combined in pairs to improve the flow of the text, avoid repetition, and make the information easier to read.

Materials and Methods
Line 150: Define JBI. For example, Joanna Briggs Institute (JBI) methodology. 
Line 169: Can use JBI, since it is described in line 150. 
ANSWER: 
Thank you for bringing this to our attention.
ACTION TAKEN:
The definition of "JBI" as "Joanna Briggs Institute (JBI)" has been added in line 145. From this point onward, the acronym "JBI" is used throughout the text, as suggested by the reviewer.

Line 210: Change was to were. 
ANSWER: Thank you for pointing that out.
ACTION TAKEN:
The change from "was" to "were" has been made, as per the reviewer's suggestion.

Line 243: Citation format is inconsistent with other citations in the document. 
ANSWER: Thank you for the suggestion.
ACTION TAKEN:
The citation has been adjusted to align with the format used consistently throughout the document.

Line 248: Spell out 4 for consistency. 
Figure 1: In identification box, there needs to be spaces after MEDLINE with Full Text (1286), Scopus (1631), Cochrane Library (355), and MedNar (510). There are inconsistencies
throughout the Figure with spacing when reporting results. For example, (n = 47) and (n=23). Revise Figure for consistency. 
ANSWER: Thank you for noting this; we have made the corresponding adjustments.
ACTION TAKEN:
The numeral "4" has been spelled out as "four" for consistency.
Figure 1:
Action taken: In the identification box, spaces have been added after "MEDLINE with Full Text (1286)", "Scopus (1631)", "Cochrane Library (355)", and "MedNar (510)", ensuring uniformity. Additionally, inconsistencies in the use of spacing when reporting results (e.g., "(n = 47)" vs. "(n=23)") have been addressed. The format "(n=X)" has been applied consistently throughout the figure.

Results
Line 285: Replace forty-seven with 47. 
Line 332: Punctuation after the parenthesis. For example, “…] (see Appendix C).”
Line 539: Instead of spelling out fifteen, provide the number (i.e., 15) for consistency in the sentence. 
ANSWER: Thank you for pointing out the inaccuracies. 
ACTION TAKEN: 
The changes were made as suggested.

Discussion
Line 558: Replace forty-seven with 47. 
ANSWER: We thank the reviewer for the suggestion made. 
ACTION TAKEN:
The change was made as suggested.

Line 568: Can use the acronym since it was previously described in the manuscript. 
ANSWER: We thank the reviewer for the suggestion made
ACTION TAKEN:
“Treatment as usual” has been replaced by TAU.

Lines 591-597: Move information up into the previous paragraph. 
ANSWER: We are thankful for your constructive remark.
ACTION TAKEN:
The two paragraphs have been merged into one, presenting the content in a more cohesive manner.

Line 598: Begin new paragraph. 
ANSWER: Your suggestion is greatly valued, and we have incorporated it.
ACTION TAKEN:
The paragraph 'In terms of interventions, cognitive behavioural … synergistically address the diverse needs of patients.' was split into two: 'In terms of interventions, cognitive behavioural … as they address both cognitive and behavioural aspects [103,109]. These data are consistent … address the diverse needs of patients.' – This division creates two clear blocks of text with distinct focuses, making it easier to read.

Lines 638-644: Move this paragraph up to the previous paragraph. 
ANSWER: Thank you for your input; it has helped improve the manuscript.
ACTION TAKEN:
The paragraph “At the educational level, interventions aim to improve knowledge about the disease and its treatment, both for patients and their carers, to facilitate treatment adherence and promote a supportive environment [54-56,62,72-73,75,77,79,81,84-85,88-90,92-94,96]. Finally, at the social level, the aim is to strengthen family and social relationships, improve communication and mutual support, and reduce the emotionality expressed in family interactions, which is crucial for successful recovery and maintenance of mental health [54,57-59,72,74,82,91,93-94].” was incorporated into the previous paragraph, as it improves the continuity of the text.

Lines 694-709: Combine into one paragraph as this information pertains to interventions. 
ANSWER: We are thankful for your constructive remark.
ACTION TAKEN: 
The two paragraphs discussing assessment and implementation contexts have been combined into a single paragraph, as the information pertains to interventions, to improve coherence and flow as requested.

Line 701: Appears information is missing from the sentence. 
ANSWER: Thank you for highlighting this important aspect.
ACTION TAKEN: 
The sentence has been revised to provide additional clarity and detail regarding the importance of continuing to explore assessment tools and their impact on clinical practice (It is underlined in green.). 

Lines 729-732: Move to above paragraph. 
ANSWER: Thank you for the suggestions.
ACTION TAKEN: 
The paragraph “Mental health nurses…equity of access to these services.” was moved to the previous paragraph, as indicated, to improve the flow of the text and the cohesion between the ideas.
4o mini

Line 736: Consider changing the word reviewers to author
ANSWER: Thank you for your observation; it has been addressed.
ACTION TAKEN: 
I followed the suggestions and changed it to "author.

Appendix B
Jackson citation: Change comma to period for consistency. 
Vidarsdottir citation: Change comma to period after al. for consistency. 
Chien citation: Remove period after et. and add space. 
Krarup Get citation: I think it should read Krarup Get T, et al. to match citation format in the Table. 
Reininghaus citation: Missing s on Netherlands. 
ANSWER: Thank you for bringing this to our attention.
ACTION TAKEN: All the suggested changes to the bibliography have been made.

Overall feedback: Throughout the table in the objectives section, sometimes acronyms were used and sometimes they were not (sometimes they were described and sometimes they were not). For clarify and ease for the reader, I suggest providing a footnote of the acronyms used (i.e., CBT) to avoid having to describe each of them throughout the table. 
ANSWER: We value your suggestion and have updated the text as recommended.
ACTION TAKEN:
The legend has been added after the table. 

Concerning the following acronyms, given that they were used multiple times throughout the article, it was decided to remove the long form after using acromion for the first time. Changes are marked in green:
Line 350 for CCT - compensatory cognitive training
line 297 for CBT - cognitive behavioural therapy 
Line 378 for COPE - cognitively orientated psychotherapy for early psychosis
Line 402 for CR/ and line 308 for CRT - Cognitive Remediation/ Cognitive Remediation Therapy
Line 300 for FEP - first-episode psychosis
Line 385 for FMSG - Family-Led Mutual Support Group
Line 408 for NEAR programme Neurocognitive Educational Approach to Remediation
Line 495 for PANSS - Positive and Negative Syndrome Scale
Lih 410 for SCIT - Social Cognition and Interaction Training
Line 417 for SM - symptom management
Line 298 for TAU - treatment as usual

Appendix C
[52] Early psychosis: A space is missing between 34 and years.
[53]: Remove extra spacing between [53] and FEP.
[54]: Remove space between / and family.
[56]: For consistency, lowercase family.
[59]: Remove extra space between ; and (.
[63]: Remove extra space between patient and /.
[66]: Add space between 35 and years.
[67]: Add space between 45 and years.
[75]: Remove extra space between risk and for.
[81]: Remove extra space between ; and [.
[86]: Lower case family for consistency.
[89]: Lower case family for consistency.
[90]: Remove extra space between / and family.
[94]: Remove extra space between [ and FEP. 
ANSWER: We value your suggestions and have updated the text as recommended.
ACTION TAKEN:
All the suggested changes have been made to improve consistency.

Overall feedback: A footnote explaining the elements of the table would help clarify the information for the reader. Additionally, it would help the reader understand the table if the table header format mirrored how the information was displayed. For example, Diagnosis/Age/Threat could read as Early psychosis/17-34 years/Patient OR Diagnosis; Age; Threat, which matches the current format. 
ANSWER: Thank you for your input; it has helped improve the manuscript.
ACTION TAKEN:
In line 7, a footnote was added ‘- Description of participant characterisation, in terms of diagnosis, age, and target.’
The table header format is used throughout the table to make it easier to read.

Appendix D
Overall feedback: The content provided is detailed and thorough. However, some minor revisions are needed.
In the implementation context column, ensure the first word after the bullet is capitalized for consistency. 
ANSWER: Thank you for your observation; it has been addressed.
ACTION TAKEN:
Some adjustments were made such as spacing, alignment and the implementation context column, the first word of each row was capitalised.

Appendix E
Overall feedback: The content provided is detailed and thorough. However, attention to formatting and how the information is presented is needed. 
For consistency, lower case name and remove uni (since multicomponent is used throughout the manuscript). 
Throughout the table, some of the modalities are capitalized and some are not, and some are explained with a dash and some with a parenthesis. For consistency purposes, please select how you want to visually display this information and make the necessary revisions throughout the table (right column).
I was not able to identify why some of the modalities were bold and some were not. Please clarify. Perhaps a footnote explaining would help clarify for the reader.
In the intervention objective descriptions, some of the descriptions end with a period and some do not. For consistency, I suggest having all descriptions end with a period.

ANSWER: Your feedback is much appreciated and has been taken into account.
ACTION TAKEN:
The different items marked were analysed to make the table clearer and easier to read. The bolt has been removed from some interventions and an attempt has been made to standardise each column, as suggested.

Appendix F
Overall feedback: Information is presented in a clear and concise manner. However, there are some errors with formatting. For example, the Services & Resources and Other categories are missing the brackets around the article numbers. In the Services & Resources category, the EPQ alignment is off. Also, capitalization of words for some of the scales is missing (i.e., Level of expressed emotion should read as Level of Expressed Emotion (LEE)). Please revise table for consistency.

ANSWER: We appreciate your detailed review and have acted on your comment.
ACTION TAKEN:
All the suggested changes have been made. The table's formatting has been revised to improve coherence.

Reviewer 2 Report

Comments and Suggestions for Authors

Dear Author,

It is a very important study in that it comprehensively summarizes the implementation status of previous research on early intervention programs for psychiatric disorders.

I have some comments and recommendations:

1. Program name/intervention goal

The program is described as being divided into single and multiple components. The programs are described in parallel, and there are many overlapping programs, giving the impression that they are not organized. How about reorganizing the explanation of the program characteristics according to the intervention goal and target?

Also, the contents described in " 3.3.1. Programme name/Intervention objective" and " 3.3.3. Intervention type - Strategy/content" are quite similar. How about integrating them?

For example, program characteristics might be categorized by target audience (patients, caregivers, patients and caregivers).

Alternatively, characteristics may be categorized according to the program's intervention objectives or goals.

Examples

・Cognitive function

・Motivation

・Social function

・Improvement of adaptation and coping skills

・Understanding and self-management of symptoms

・Strengthening family function, reducing the burden on caregivers

・Personality growth and recovery using group psychotherapy

2. Frequency

The number of sessions, treatment period, and session frequency are shown in the range from minimum to maximum. In the 47 studies included in your research, how about showing the median number of sessions, treatment duration, and session frequency? If possible, showing the variation in frequency of each item in a graph or other format may make it easier for readers to understand.

3. Subjects

- Three studies included the age group of 65 years [64,76,81]. Does this mean late-onset schizophrenia? Can it be called FEP?

4. Discussion and Conclusion

It is suggested that " Mental health nurses, already trained in a wide range of evidence-based interventions, should be encouraged to further engage in early intervention practices within multidisciplinary teams. " The discussion does not discuss the importance of mental health nurses' involvement in intervention programmes of FEP in comparison with other professions. Even when making such a proposal, we recommend that the discussion discuss the importance of mental health nurses' involvement in intervention programmes of FEP in light of the findings of the target 47 studies and other previous studies.

Author Response

Response to Reviewer 2 Comments
We would like to express our sincere thanks for your attention and dedication in reviewing our article. Your observations and suggestions were extremely valuable and contributed significantly to improving the quality of the work. Thank you for your time, effort and constructive approach, which undoubtedly helped to make the article clearer and more complete.
Comments 1: Program name/intervention goal
The program is described as being divided into single and multiple components. The programs are described in parallel, and there are many overlapping programs, giving the impression that they are not organized. How about reorganizing the explanation of the program characteristics according to the intervention goal and target?
Also, the contents described in " 3.3.1. Programme name/Intervention objective" and " 3.3.3. Intervention type - Strategy/content" are quite similar. How about integrating them?
For example, program characteristics might be categorized by target audience (patients, caregivers, patients and caregivers).
Alternatively, characteristics may be categorized according to the program's intervention objectives or goals.
Examples
・Cognitive function
・Motivation
・Social function
・Improvement of adaptation and coping skills
・Understanding and self-management of symptoms
・Strengthening family function, reducing the burden on caregivers
・Personality growth and recovery using group psychotherapy

Response 1 
We would like to thank you very much for your comments and suggestions. We recognise that organising and clearly presenting the large amount of information contained in our manuscript is fundamental to its comprehension and impact. We therefore appreciate the opportunity to revise our approach in the light of your comments.

Number and organisation of programes
Our study is a scoping review, the main aim of which is to extensively map the existing literature on intervention programmes. By nature, this type of review seeks to capture the diversity and complexity of a field, including aspects that may seem overlapping or redundant. This comprehensive approach is intentional and necessary to ensure that all the nuances of the programmes are considered.
We recognise, however, that the parallel presentation of programmes could give the impression of a lack of organisation. In response to this concern, we have revised the text in section 3.3.1 to clarify the logic of the structure adopted. We have added the following explanation:
"Given the high number of articles, an effort was made to group them into single-component interventions (e.g., CBT, computerised interventions, cognitive remediation, psychoeducation) or those that integrate multiple components (e.g., CBT + CM + psychoeducation). This categorisation was adopted for two primary reasons: to facilitate the organisation and analysis of the articles and to highlight the programme characteristics. The classification provides a clear framework for presenting programmes with varying levels of complexity and allows us to underscore the depth of focused interventions and the breadth of integrated programmes, without overwhelming the reader with unstructured details.
The included studies encompassed different psychosocial interventions, reflecting a wide array of cognitive, behavioural, and social approaches tailored to varying objectives and characteristics. It is acknowledged that some single-component interventions may be part of broader programmes; however, they were analysed independently when the study’s primary focus was on a single component, as specified in the original articles. Appendix E provides a brief overview of the different programmes, including their names (if assigned), intervention objectives, and whether they are single- or multi-component. Note that there may be selected articles in which only a single isolated intervention is analysed, which might be part of a broader programme (e.g., [88])."
About objectives and strategies/content
Thank you for your comment on the similarity between the sections ‘3.3.1 Name of the programme/Objective of the intervention’ and ‘3.3.3 Type of intervention - Strategy/Content’. This redundancy was noticed due to the thematic proximity between the two sections. We have kept these sections separate, as they deal with different aspects:
•    Objectives: refer to the main goals of the interventions, such as improving cognitive function, promoting coping skills or reducing the burden on carers.
•    Strategies and content: include the specific methods and tools used to achieve these objectives, such as training sessions, the use of technology or group activities. This distinction has been preserved to ensure that both the ‘ends’ and the ‘means’ are clearly communicated to the reader. We have revised the text of these sections to make the conceptual difference between the two topics more explicit, to avoid confusion.

About organising by target group or intervention objectives
We carefully considered the suggestion of grouping programmes by target audience or objectives. We have identified practical limitations to this approach:
•    Multicomponent programmes: Many programmes include interventions aimed at both patients and carers, making it difficult to categorise without losing important nuances.
•    Diversity of goals: Some programmes have broad, interrelated goals (e.g. promoting motivation and coping skills simultaneously), which would make rigid categorisation impractical. Despite these limitations, the new introduction in section 3.3.1 (cited above) seeks to justify the logic of the categorisation adopted and provide a clear context for the organisation of the data presented.

ACTION TAKEN:
Based on the feedback received, we have made the following improvements:
1. We have included an explicit explanation in section 3.3.1 about the categorisation into uni- and multi-component interventions, as mentioned above.
2.    We have revised the text of sections 3.3.1 and 3.3.3 to make explicit the difference between objectives and strategies/content, maintaining the distinction between these two aspects.
3.    We have revised the general text to improve clarity and accessibility, ensuring that the presentation of the data remains aligned with the exploratory objectives of the scoping review.

Comments 2. Frequency
The number of sessions, treatment period, and session frequency are shown in the range from minimum to maximum. In the 47 studies included in your research, how about showing the median number of sessions, treatment duration, and session frequency? If possible, showing the variation in frequency of each item in a graph or other format may make it easier for readers to understand.
Response: 
Thank you very much for your comment and constructive suggestion. We agree that a more detailed presentation of the variation in the variables of number of sessions, duration of treatment and frequency of sessions could make the data easier to understand. As this is a scoping review, whose main aim is to broadly map the characteristics and diversity of interventions in the literature, we have chosen to provide an overview of the data, without going into specific details such as the calculation of the median. The great heterogeneity in the data reported, as well as the lack of some data in several studies, made it difficult to reliably calculate the median for all variables.
In response to your suggestion, we have revised section 3.3.2 to make the variation observed in each variable clearer, highlighting the amplitudes and extreme values. In this way, we still offer a comprehensive view of therapeutic practices, in line with the objectives of the scoping review.
Thank you again for your valuable feedback and for allowing us to improve the explanation.
Comments 3. Subjects
- Three studies included the age group of 65 years [64,76,81]. Does this mean late-onset schizophrenia? Can it be called FEP?
 RESPONSE: 
We deeply appreciate your comment about the inclusion of participants aged up to 65 in the studies analysed. We would like to clarify, based on the studies mentioned, that the age range does not necessarily imply a diagnosis of late-onset schizophrenia, but rather the inclusion of individuals in the early stages of psychosis, according to the definition of first psychotic episode (FEP).
Review [64]: The study in question includes participants aged between 18 and 65. Although the age range extends up to 65, the inclusion criteria were clearly defined for individuals with a primary diagnosis of psychotic spectrum disorder according to DSM-IV. These participants had a maximum of two episodes of psychosis or up to two years of adequate treatment for psychosis. Therefore, the participants are in the initial phase of the illness, within the definition of FEP. The study does not mention or focus on late onset schizophrenia, and the wider age range was a practical consideration to include a broader spectrum of participants in the early stage of psychosis. Thus, the sample remains in line with the definition of FEP and is appropriate for our review.
Review [76]: This study involves participants with a first psychotic episode (FEP) and an elevated risk for psychosis (UHR), aged between 18 and 65. Although the age range covers up to 65 years, the focus is on individuals who are at the beginning of their psychotic trajectory or are at increased risk of developing psychosis. In this way, we are not talking about late onset schizophrenia, but psychosis in the early stages. We consider that, based on the inclusion criteria, which guarantee relevance to FEP, the inclusion of this study is completely appropriate for our review.
Review [81]: The study recruited participants aged between 16 and 65 from Greater Manchester Mental Health NHS Foundation Trust's early psychosis intervention services. The inclusion of participants from the age of 16 aligns with the definition of FEP, as these individuals are in the early stages of psychosis or are at high risk for a first psychotic episode. The study focuses on early-stage psychosis and not late-onset schizophrenia. Therefore, this study is also appropriate for our review, as it focuses on FEP, according to the defined inclusion criteria.
In summary, although the age range of some studies extends up to 65 years, they all focus on individuals with a first psychotic episode or high risk for psychosis, and not on late onset schizophrenia. 
Thank you again for your comment and we believe that, with the above explanations, the studies are properly contextualised and relevant to our review.
Comments 4. Discussion and Conclusion

It is suggested that " Mental health nurses, already trained in a wide range of evidence-based interventions, should be encouraged to further engage in early intervention practices within multidisciplinary teams. " The discussion does not discuss the importance of mental health nurses' involvement in intervention programmes of FEP in comparison with other professions. Even when making such a proposal, we recommend that the discussion discuss the importance of mental health nurses' involvement in intervention programmes of FEP in light of the findings of the target 47 studies and other previous studies.

RESPONSE: 
Thank you for your detailed comment and we recognise the relevance of your observation. We agree that the initial discussion did not adequately address the importance of mental health nurses' involvement in early intervention in psychosis programmes compared to other professions, as suggested. After reflecting on the point raised, we recognised that the previous conclusions could give the impression that we were jumping to conclusions about the specific role of nurses, without adequate support from the findings of the 47 included reviews.

Based on this assessment, we have revised the manuscript to reflect a more balanced approach. In the revised text, we emphasise the interdisciplinary role of mental health teams and highlight that nurses, as well as other professionals, have specific contributions to make in promoting a holistic and inclusive approach to care. Although we briefly mentioned the possibility of developing specific programmes for nurses, it is now clear that this is a potential recommendation for future exploration, rather than a conclusion based on the findings of this review.

Thank you again for your feedback, which has helped us to improve the clarity and accuracy of the manuscript.

Round 2

Reviewer 2 Report

Comments and Suggestions for Authors

Dear Author

I understand that the study objective is to describe the diversity and complexity of intervention programs for first-episode psychology in as much detail as possible. Therefore, I understand that it is appropriate to describe various interventions under the categorization of single and multiple interventions.

In addition, the explanation of the concept of the intervention's purpose and strategy/content was added, and I was able to clearly understand the difference.

Regarding frequency, I understand that an overview was provided because of heterogeneity and lack of data. There was a comment below, but I could not find any evidence that it was revised in 3.3.2. Please check again for the revisions.

"In response to your suggestion, we have revised section 3.3.2 to make the variation observed in each variable clearer, highlighting the amplitudes and extreme values. In this way, we still offer a comprehensive view of therapeutic practices, in line with the objectives of the scoping review."

I also confirmed that the focus was not on late onset schizophrenia and that the papers on FEP were appropriately reviewed.

I also acknowledged that you mentioned the role of mental health nurses in Future research.

Author Response

Dear Reviewer,
We sincerely thank you for your valuable feedback. We consider it essential to carefully address every suggestion we receive.

In line with the nature of a scoping review, our main objective was to map and describe the characteristics of interventions rather than to provide a quantitative synthesis of the data. The heterogeneity across studies, coupled with incomplete data and the complexity of multiple interventions in some programs, made it impractical to calculate consistent and valid measures, such as medians or variations.
However, in response to your suggestion, we have made efforts to report ranges and extreme values wherever possible to illustrate the diversity of the data. For each variable—number of sessions (NS), treatment duration (TD), session frequency (FS), and follow-up (FU)—we described the variation from minimum to maximum values, as detailed in lines 451–457.
Previously, we stated:
"In response to your suggestion, we have revised section 3.3.2 to make the variation observed in each variable clearer, highlighting the amplitudes and extreme values. In this way, we still offer a comprehensive view of therapeutic practices, in line with the objectives of the scoping review."
Upon reflection, we realized that no substantial changes had been made, given the inherent limitations of this type of review. However, we now recognize the importance of including a clear rationale. Consequently, we added the following statement on lines 458–460:
"Due to the large heterogeneity of the values for number of sessions (NS), treatment duration (TD), session frequency (FS), and follow-up (FU), the characteristics were described with a presentation of their amplitudes (minimum and maximum values) to illustrate the variation in the data, where possible."

We are grateful for the opportunity to clarify this point and hope the updated text meets your expectations.
Additionally, we appreciate your positive acknowledgment of the explanation of the intervention's purpose and strategy/content, as well as your recognition of our focus on first-episode psychosis (FEP) and the appropriate exclusion of studies on late-onset schizophrenia. Finally, we note your acknowledgment of the role of mental health nurses as an avenue for future research.
Thank you again for your valuable feedback, which has helped us to improve the manuscript.